# Transcriptome Analysis Reveals Potential Mechanisms for Ethylene-Inducible Pedicel–Fruit Abscission Zone Activation in Non-Climacteric Sweet Cherry (*Prunus avium* L.)

Seanna Hewitt [1,†], Benjamin Kilian [2,†], Tyson Koepke [3], Jonathan Abarca [1], Matthew Whiting [4] and Amit Dhingra [1,*]

1 Department of Horticulture, Washington State University, Pullman, WA 99163, USA; seanna.hewitt@wsu.edu (S.H.); j.abarca1135@gmail.com (J.A.)
2 Department of Agriculture, African Christian University, Lusaka H985+XQ3, Zambia; kilianbe@gmail.com
3 Department of Crop and Soil Sciences, Washington State University, Pullman, WA 99163, USA; tkoepke@wsu.edu
4 Irrigated Agriculture Research and Extension Center, Washington State University, Prosser, WA 99350, USA; mdwhiting@wsu.edu
* Correspondence: adhingra@wsu.edu
† Authors contributed equally to this work.

**Abstract:** The harvesting of sweet cherry (*Prunus avium* L.) fruit is a labor-intensive process. The mechanical harvesting of sweet cherry fruit is feasible; however, it is dependent on the formation of an abscission zone at the fruit–pedicel junction. The natural propensity for pedicel—fruit abscission zone (PFAZ) activation varies by cultivar, and the general molecular basis for PFAZ activation is not well characterized. In this study, ethylene-inducible change in pedicel fruit retention force (PFRF) was recorded in a developmental time-course with a concomitant analysis of the PFAZ transcriptome from three sweet cherry cultivars. In 'Skeena', mean PFRF for both control and treatment fruit dropped below the 0.40 kg-force (3.92 N) threshold for mechanical harvesting, indicating the activation of a discrete PFAZ. In 'Bing', mean PFRF for both control and treatment groups decreased over time. However, a mean PFRF conducive to mechanical harvesting was achieved only in the ethylene-treated fruit. While in 'Chelan' the mean PFRF of the control and treatment groups did not meet the threshold required for efficient mechanical harvesting. Transcriptome analysis of the PFAZ region followed by the functional annotation, differential expression analysis, and gene ontology (GO) enrichment analyses of the data facilitated the identification of phytohormone-responsive and abscission-related transcripts, as well as processes that exhibited differential expression and enrichment in a cultivar-dependent manner over the developmental time-course. Additionally, read alignment-based variant calling revealed several short variants in differentially expressed genes, associated with enriched gene ontologies and associated metabolic processes, lending potential insight into the genetic basis for different abscission responses between the cultivars. These results provide genetic targets for the induction or inhibition of PFAZ activation, depending on the desire to harvest the fruit with or without the stem attached. Understanding the genetic mechanisms underlying the development of the PFAZ will inform future cultivar development while laying a foundation for mechanized sweet cherry harvest.

**Keywords:** ethephon; ethylene; auxin; RNA-seq; transcriptome; *Prunus avium*; abscission; Rosaceae; mechanical harvest

## 1. Introduction

Sweet cherry (*Prunus avium* L.), a member of the Rosaceae family, is a commercially important tree fruit crop throughout the world, with approximately 2.3 million tons produced annually [1,2]. In addition to its worldwide economic value, production and consumption of sweet cherries has increased in recent years as consumers have become aware of

their nutritional benefits [2]. The industry is evolving to address the growing market and the availability of labor during harvest and has begun adapting harvesting strategies to meet new consumer demand for stemless cherries [3,4]. Traditional harvesting methods focus on separating the fruit from the tree at the pedicel–peduncle junction, leaving the pedicel (stem) attached to the fruit [5]. Mechanical harvesting, on the other hand, is best achieved when the fruit abscises easily at the fruit–pedicel junction [4]. Increasing labor costs associated with traditional hand harvesting, in addition to the growing demand for stemless fruit, has made the adoption of mechanical harvesting strategies attractive if they can be made uniform across cultivars [6]. However, the harvesting of sweet cherries has presented a unique set of challenges. Unlike sour cherries (*Prunus cerasus*), which develop an anatomically and histologically distinct fruit–pedicel abscission zone (PFAZ) and separate with ease, sweet cherry cultivars display phenotypic differences in PFAZ activation and consequent ease of fruit separation—some cultivars require excessive force to separate fruit at the PFAZ, which tends to compromise fruit quality and integrity [6,7].

While sweet cherry and peach belong to the same sub-family, the former bears non-climacteric fruits that do not produce ethylene autocatalytically during ripening and senescence. This is most likely due to the presence of several stop codon mutations in the ethylene biosynthesis and perception genes in sweet cherry [8]. However, sweet cherry fruit from some cultivars exhibit a novel developmental response to exogenous ethylene application [9]. Exogenous ethylene application can induce or enhance PFAZ activation, loosening the fruit and facilitating efficient mechanical harvesting [5,10].

Pedicel–fruit retention force (PFRF) is used as a direct measure of PFAZ activation and serves as a metric to determine the mechanical harvestability of the fruit. An average PFRF value of 0.40 kg-force (3.92 N) is considered the threshold for mechanical harvestability, though this will depend on the actuation method of the harvester [6,10,11]. In sweet cherry, a reduction in PFRF can be induced through the application of ethephon (2-chloroethylephosphonic acid), a commercially available plant growth regulator that is rapidly metabolized to ethylene [6,10]. In both sweet and sour cherry, ethephon has been demonstrated to be a viable option for reducing the threshold for mechanical harvest without negatively impacting fruit quality, when applied in appropriate concentrations; however, high rates of ethephon have been associated with reduction in fruit quality (e.g., gummosis, defoliation, lenticel enlargement), and thus, cultivars with naturally low PFRF are desired [6,10–12]. The natural PFRF of sweet cherry varies by cultivar, with some varieties, like 'Skeena', exhibiting an auto-abscising phenotype and requiring no exogenous ethylene to induce abscission. Representing an intermediate phenotype, 'Bing' can be induced to abscise when ethephon is applied approximately 14 days before harvest [6,10]. 'Chelan', on the contrary, does not abscise naturally or in the presence of ethephon [10].

Sweet cherry phenotyping studies have shown that PFRF values remain consistent for these cultivars across multiple years, indicating that the abscission phenotypes are genetically stable and can perhaps be manipulated at the genetic level [6]. Furthermore, these findings suggest that the standardization of PFRF for mechanical harvesting across cultivars is possible if the ideal ethephon or other treatment regimens are determined for individual sweet cherry cultivars. Despite the extensive physiological characterization of PFAZ integrity across sweet cherry varieties, the underlying molecular basis for PFAZ structural differences has not been previously elucidated.

As understood from studies in model plant systems, such as Arabidopsis and tomato, abscission at the fruit–pedicel junction entails a series of hallmark structural changes: the middle lamella is dissolved by hydrolytic enzymes, such as polygalacturonase and cellulase [13]; cell walls in the separation layer thicken, and cell wall components become hydrated as a result [14]; primary cell walls break down as abscission progresses, resulting in the formation of large intercellular cavities [15,16], and lignin deposits accumulate proximally to the abscission zone, forming part of a peridermal boundary layer that will serve to protect the pedicel scar following fruit separation [16–19]. This process is thought to operate in a similar manner in fruit crops including apple, peach, and olive, although

species and cultivar-specific differences, particularly with regards to the chemical induction of abscission in these crops, are not yet well understood [20–24].

According to the current model of abscission in plants, the genetic events underlying cellular structure modification at the PFAZ largely involve an interplay between ethylene- and auxin-associated pathways [18,25–28]. The binding of ethylene to corresponding receptors (ETRs) in PFAZ cells initiates signal transduction pathways, leading to the activation of numerous, ethylene-responsive transcription factors (ERFs), which elicit different modulatory roles. The ethylene response and signaling network ultimately results in the initiation of cell death, a reduction in cell wall adhesion, and the separation of the fruit from the pedicel [18,29]. Working antagonistically to ethylene is the phytohormone auxin. Auxin's biologically active form, free indole-3-acetic acid (IAA), decreases the sensitivity of plant organs to ethylene. The genetic and metabolic factors governing auxin homeostasis ensure that an appropriate balance is maintained between free IAA, IAA-conjugates, and auxin degradation during different developmental stages [25,30]. The presence of higher levels of free IAA corresponds to inhibited or delayed ethylene-dependent developmental responses like abscission zone formation and/or activation [31]. Decreasing polar auxin transport across the abscission zone in sweet cherries by girdling methods results in increased fruit abscission [32]. Additionally, in grape, the application of inhibitors of auxin transport has been shown to promote increased abscission [33].

While there is evidence implicating the involvement of numerous ethylene-associated transcription factors and auxin-associated genes in the modulation of PFAZ activation, species-specific modes of action have yet to be resolved [29]. Furthermore, mechanisms involving the transmission of hormonal and other signals upstream of AZ activation, as well as specifics of the enzyme-driven remodeling and redeposition of cell wall constituents in sweet cherry, are in need of further elucidation. The stimulation of PFAZ activation in response to exogenous ethylene application in the non-climacteric fruit sweet cherry represents a unique biological system to elucidate the process of inducible abscission. An improved understanding of the interplay between hormone response, signaling, and the activation of abscission-associated genes and pathways in sweet cherry will facilitate improved strategies for planned induction or the inhibition of PFAZ activation.

To elucidate the molecular bases for differences in abscission phenotypes among sweet cherry cultivars and to correlate this information with fruit development, time course physiological measurements of the PFRF along with concomitant transcriptome analysis of the PFAZ tissue from ethylene-treated and control 'Bing', 'Skeena', and 'Chelan' were conducted. The hypothesis that cultivar-specific gene expression differences in ethylene- and auxin-responsive pathways are directly correlated to the differences in abscission phenotypes was evaluated. Additionally, this work aimed to reveal other genes and enriched gene ontologies associated with key metabolic processes involved in pedicel–fruit abscission in sweet cherry. The results of this study provide potential genetic targets for PFAZ activation in sweet cherry, which are expected to inform strategies for improving PFAZ phenotypes conducive to different harvesting approaches.

## 2. Materials and Methods

### 2.1. Plant Material

The sweet cherry trees used in this study are located at Washington State University's Roza Farm, 10 km north of Prosser, Washington, USA ($46.2°$ N, $119.7°$). Trees were irrigated weekly from bloom to leaf senescence with low-volume, under-tree micro-sprinklers and grown using standard orchard management practices. Trees had an in-row spacing of 2.44 m (8 ft) and between-row spacing of 4.27 m (14 ft). Rows were planted in a north–south orientation and trained to a Y-trellis architecture.

### 2.2. Ethephon Application

Ethephon (formula 240 g/L [2 lbs/gal]) was applied via air-blast sprayer at 3.5 L ha$^{-1}$ (3 pt A$^{-1}$) with a 1871 L ha$^{-1}$ (200 g A$^{-1}$) spray volume [10]. Ethephon applications

and measurements were conducted in three different years (2010, 2013, and 2014). Each replication was performed in the same orchard block, using distinct trees within the block. Treatment application was done early in the morning (between 0600 and 0800 h) to reduce the effects of ethylene evolution from warm temperatures and wind, as previously described [10].

Optimal ethephon application time for 'Bing' had been established previously as 14 days before harvest (DBH) [10] or 80% fruit maturation. Because 'Bing', 'Chelan', and 'Skeena' have different timelines for the maturation of fruit after bloom, ethephon was applied at ca. 80% maturation for each of the cultivars in the 2014 growing season. This percentage coincided with 14 DBH for 'Bing', 12 DBH for 'Chelan', and 16 DBH for 'Skeena' (Figure S1). Information regarding ethephon treatment, and PFRF results for 2010 and 2013 can be found in Supplementary Materials Data S2.

### 2.3. PFRF Measurements, Color Evaluation, and Abscission Zone Sampling

In all three years, sampling and measurements were conducted at the following time points for each sweet cherry cultivar: (1) immediately prior to the application of ethephon or $H_2O$, (2) 6 h after the application of ethephon or $H_2O$, and (3) at harvest.

At each sampling time, ten fruit were randomly selected for analysis from each of the four trees/cultivar/treatment. PFRF was measured using a modified digital force gauge (Imada). Each fruit was manually categorized by exocarp color based on a 1–7 scale developed specifically for sweet cherry by CTIFL (Centre technique interprofessionnel des fruit et legumes, France) (Supplementary Materials Data S3) [34]. In addition to the collection of PFRF values, the abscission zones of 10 fruit, representing ten biological replicates from each cultivar/treatment, were harvested from corresponding trees at each time point per the following steps: (1) Using a new razor blade, the fruit was first cut approximately 0.5 cm below the pedicel, leaving the pedicel and a thin disc of fruit/skin attached; (2) Two sets of parallel cuts were made downward on the cherry fruit disc on either side of the stem, effectively making a cubic piece of fruit 3 mm × 3 mm × 3 mm attached to the pedicel; (3) The pedicel was cut off directly above the fruit and the cube of fruit tissue, consisting of the abscission zone and some fruit and pedicel tissue was placed in a 15 mL falcon tube and flash-frozen for subsequent processing (Figure S4).

### 2.4. Total RNA Extraction

Excised sweet cherry abscission-zone-region tissue derived from ten fruits representing ten biological replicates/cultivar/time point was pooled, pulverized and homogenized into a single sample using a SPEX SamplePrep FreezerMill 6870 (Metuchen, NJ, USA) and transferred to storage at −80 °C. Total RNA was extracted using an acid guanidinium thiocyanate phenol chloroform extraction method similar to that previously described [35]. Briefly, 1 mL of 0.8 M guanidinium thiocyanate, 0.4 M ammonium thiocyanate, 0.1 M sodium acetate (pH 5.0, 5% w/v glycerol), and 38% v/v water-saturated phenol were added to approximately 100 mg powdered tissue, shaken to evenly mix the sample and incubated at room temperature (RT) for 5 min. Then, 200μL chloroform was added and shaken vigorously until the entire sample became homogenously cloudy and then was incubated (RT, 3 min). Samples were centrifuged at 17,000× *g* at 4 °C for 15 min, and the aqueous upper phase was transferred to a sterile 1.5 mL microcentrifuge tube. To this, 600μL 2-propanol was added, inverted 5–6 times, and incubated at RT for 10 min. Samples were centrifuged at 17,000× *g* at 4 °C for 10 min, and the supernatant decanted. Then, 1 mL 75% DEPC ethanol was added to the pellet, vortexed for 10 s, and centrifuged at 9500× *g* at 4 °C for 5 min. Pellets were suspended in RNase free water and incubated at 37 °C with RNase free *DNaseI* for 30 min, which was subsequently inactivated (65 °C, 10 min).

Extracted RNA was quantified, and its quality was checked using the Bio-Rad (Hercules, CA, USA) Experion system using the Experion RNA High Sensitivity Analysis kit or the Agilent (Santa Clara, CA, USA) 2100 Bioanalyzer system using the RNA NanoChip and Plant RNA Nano Assay Class.

### 2.5. RNA Sequencing and Assembly

RNA samples that passed the quality threshold of RIN 8.0 were sequenced at the Michigan State University Genomics Service Center for library preparation and sequencing. The Illumina Hi Seq 2000 sequencing platform (San Diego, CA, USA) was used to sequence $2 \times 100$ PE reads from the cDNA libraries generated from the above RNA extractions, representing a single sample derived from 10 biological replicates of each cultivar, treatment, and time point. cDNA and final sequencing library molecules were generated with Illumina's TruSeq RNA Sample Preparation v2 kit (San Diego, CA, USA) and instructions with minor modifications. Modifications to the published protocol include a decrease in the mRNA fragmentation incubation time from 8 min to 30 s to create the final library of proper molecule size range. Additionally, an Aline Biosciences' (Woburn, MA, USA) DNA SizeSelector-I bead-based size selection system was utilized to target the final library molecules for a mean size of 450 base pairs. All libraries were quantified on a Life Technologies (Carlsbad, CA, USA) Qubit Fluorometer and qualified on an Agilent (Santa Clara, CA, USA) 2100 Bioanalyzer.

Read preprocessing and assembly were conducted in CLC Genomics Workbench (8.5.1). Briefly, RNAseq read datasets were processed with the CLC Create Sequencing QC Report tool to assess read quality. The CLC Trim Sequence process was used to trim the first 16 bases due to GC ratio variability and for a Phred score of 30. All read datasets were trimmed of ambiguous bases. Illumina reads were then processed through the CLC Merge Overlapping Pairs tool, and all reads were de novo assembled to produce contiguous sequences (contigs). A single master assembly was generated from the combined read data from the 'Bing', 'Chelan', and 'Skeena' cultivars at each time point (Supplementary Materials File S5). Assembled contigs passed the filter criteria of >200 base length combined with >2x average read coverage. The cultivar-specific, non-trimmed read sets were mapped back to the master assembly to generate individual read mappings for each cultivar, treatment, and time point. Read counts were normalized for each mapping group using the reads per kilobase per million reads (RPKM) method (Mortazavi et al., 2008). All the sequence data is available from the NCBI Short Read Archive—Bing accession: SRX2210365; Chelan accession: SRX2210366; Skeena accession: SRX2210367 submitted under BioProject PRJNA329134.

### 2.6. Differential Expression Analysis

A two-pronged approach, using Kal's Z-test and the NOIseq-sim R package, was employed to identify contigs with the highest likelihood of being differentially expressed [36–38]. Only contigs that passed established threshold filtering for both methods were considered for further analysis.

Genes with the highest probability of being differentially expressed were first identified using the NOIseq-sim package in OmicsBox, which is designed to infer the probability of differential expression by modeling biological replications in silico for RNAseq experiments [37–39]. This approach has been successfully employed for differential expression analysis in other crops, including peach and rice [40,41]. Default parameters were used to simulate five in silico replications with a simulated replicate size of 0.2 and a set variability of 0.02 in each replication.

Next, Kal's Z-tests were performed in CLC Genomics Workbench 8.5.1 to add another level of stringency to the identification of differentially expressed genes. A paired experiment comparing the read count values for ethephon treatment to control values corresponding to each cultivar was performed.

The final sequence selection was reduced to 3190 contigs, which were considered to have a high probability of being differentially expressed based on conformity to all of the following criteria for at least one treatment/time: (1) $|\log2FC| > 1$, (2) NOIseq probability of DE > 0.8, (3) Kal's Z-test FDR corrected *p*-value < 0.05 (Supplementary Materials File S6).

### 2.7. Functional Annotation and GO Enrichment Analysis

Assembled sweet cherry contigs were annotated using the Blast2GO feature in Omics-Box (version 1.2.4). Contigs were blasted for greatest sequence homology against the NCBI Viridiplantae database and subsequently assigned to their corresponding gene ontology (GO) terms as described previously [42–44].

GO enrichment analysis using Fisher's exact test was also conducted in OmicsBox to identify the cellular components, molecular functions, and biological processes that were over- or under-represented in the ethephon treated fruit at harvest in comparison with the control fruit [42]. Based on the differential expression analysis, for each sweet cherry cultivar, lists representing transcripts with NOIseq probability >0.8, Kal's Z-test FDR corrected $p$-value < 0.05, and |logFC| > 1 at the harvest time point were produced. These lists served as the treatment datasets for enrichment analyses, and the master annotated transcriptome was used as the reference dataset (Supplementary Materials File S7). The FDR-corrected $p$-value cutoff for enrichment was set to 0.05. Following separate enrichment analyses for each cultivar, enriched GO terms that were shared between cultivars or unique to a single cultivar were identified using the Venn Diagram application in OmicsBox.

### 2.8. RT-qPCR Validation

Targets for RT-qPCR validation were selected from a list of genes known to be involved in ethylene response and cell wall breakdown (Supplementary Materials File S8). Primers were designed based on the near full-length transcript sequences to amplify an approximately 100–150 bp region in the 3′ region of target transcripts. A bacterial luciferase gene was used as a spiked reference, with 50 ng added per reaction.

Library preparation, target amplification, and expression analysis were conducted in accordance with previously published methods, with minor modifications (Hendrickson et al., 2019). The Invitrogen (Waltham, MA, USA) VILO cDNA synthesis kit was used to generate three biological replicates of cDNA from three independent RNA samples isolated for each time point per manufacturer's instructions. Replicate cDNAs were then pooled into a single sample (50 ng/uL). At least 4–6 technical replicate RT-qPCR reactions were performed using iTAQ with ROX and SYBR (BioRad), and 20 μL reactions were prepared as per the manufacturer recommendations. A total of 2 μL of cDNA diluted to 50 ng/μL RNA equivalents was used per reaction with 5 μL $H_2O$, 2 μL of each primer (10 μM), and 10 μL of iTAQ SYBR Green Supermix with ROX. The RT-qPCR reactions were performed on a Stratagene MX3005 using the following parameters: 95 °C 5 min; 50 cycles of 95 °C 30 s, 57 °C 30 s, 72 °C 30 s; 72 °C 5 min. Fluorescence readings were taken at the end of each elongation step. A melting step was performed following the cycles at 95 °C for 30 s, 54 °C for 30 s and ramp up to 95 °C to produce a dissociation curve.

To capture PCR efficiency in the data, Cq values and efficiencies were calculated for each reaction using the LinRegPCR tool (Ramakers et al., 2003, Ruijter et al., 2009). Cq values resulting from efficiencies below 1.80 or above 2.20 were judged unacceptable and were treated as unsuccessful or undetected amplifications. Cq values with efficiency values that were within expected parameters but exceeded (or equaled) 40.00 were also deemed unacceptable and disregarded in downstream analysis. In the same manner, Cq values of 35.00–39.99 were determined to be of low confidence and were marked for special consideration in downstream analysis. Fold-change expression was determined from the Cq values of all gene targets (among all replicates of all samples) among the 'Bing', 'Chelan', and 'Skeena' cultivars using the Pfaffl method (Pfaffl, 2001). Expression values were determined with reference to the luciferase spiked gene (Supplementary Materials Data S8).

### 2.9. Short Variant Identification

The GATK best practices pipeline for short variant discovery was used to identify SNPs and indels in key, differentially expressed ethylene- and auxin-associated contigs, with minor modifications [45,46]. Briefly, a group of paired, untrimmed reads from each

sweet cherry cultivar was aligned to a designated reference fasta in CLC Genomics Workbench 8.5.1. The reference file contained only the sequences for the assembled contigs that had been previously assigned GO annotations in OmicsBox. The resulting three alignments were exported as BAM files for subsequent use in the GATK pipeline. A reference fasta index and dictionary were created using Samtools and Picard software programs, respectively. Within the GATK (v. 4.1.7.0) suite, the HaplotypeCaller tool was used to identify variants between parental haplotypes for each cultivar; the results from all three cultivars were then merged into a single GVCF file using the CombineGVCFs tool. Finally, the GenotypeGVCFs tool was used to perform joint genotyping on the GVCF file containing variant information for each cherry cultivar. The SelectVariants tool was used to filter out low-quality and low-confidence calls; minimum depth of coverage (DP) was set at $30\times$, and minimum quality of depth (QD), which is directly related to Phred score, was set to 30. The results were visualized and the called variants in differentially expressed genes of interest were confirmed using Integrative Genomics Viewer (v. 2.8.2).

## 3. Results and Discussion

### 3.1. Pedicel–Fruit Retention Force

The application of ethephon at 80% of fruit development [10] ensured developmental equivalency of PFRF and tissue sampling time points across cultivars. This percentage coincided with 12 DBH for 'Chelan', 14 DBH 'Bing', and 16 DBH for 'Skeena'. The PFRF values of control fruits decreased naturally over time, with reductions of 75.5%, 74.0%, and 37.5% observed for 'Skeena', 'Bing', and 'Chelan,' respectively. The application of ethephon decreased mean PFRF value in comparison with the respective controls, but whether this decrease was biologically significant and resulted in the achievement of the threshold for mechanical harvesting varied across cultivars (Figure 1).

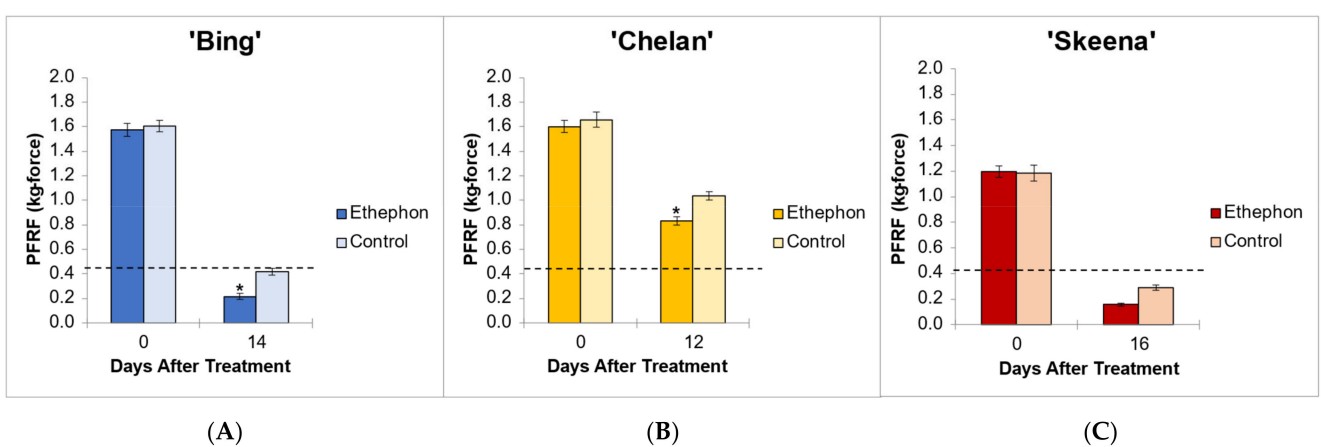

(A)  (B)  (C)

**Figure 1.** Endpoint mean control and ethephon-treatment PFRF values for (**A**) 'Bing' (**B**), 'Chelan', and (**C**) 'Skeena' PFAZ tissues. The dotted line represents the threshold PFRF for mechanical harvest. Asterisks indicate significant difference between ethephon-treated and control fruit at harvest ($p < 0.05$).

In 'Skeena', mean PFRF for both control and treatment fruit dropped below the 0.40 kg-force (3.92 N) threshold for mechanical harvesting, reaching final mean PFRF values of 0.29 kg-force (2.84 N) and 0.156 kg-force (1.53 N), respectively (Figure 1). These findings indicate that while 'Skeena' is capable of natural abscission, ethephon application does significantly increase the AZ activation response, causing PFRF values to decrease significantly compared to control fruit by the harvest date.

Similarly, for 'Bing', mean PFRF for both control and treatment groups decreased over time; however, a mean PFRF conducive to mechanical harvesting was only achieved in the ethephon-treated fruit, which reached a final PFRF value of 0.215 kg-force (2.11 N), significantly lower than the control value of 0.418 kg-force (4.1 N) (Figure 1). This suggests

that the inducibility of 'Bing' is resultant of a similar, yet less dramatic, natural decrease in PFRF that is enhanced by ethephon treatment.

'Chelan' exhibited a statistically significant PFRF response at the time of harvest in the ethephon-treated fruit in comparison with the control; however, the final PFRF of 0.832 kg-force (8.16 N) of the treatment group was not reduced to the threshold required for efficient mechanical harvesting (Figure 1). These physiological results support the observations that 'Chelan' forms neither a developmental nor an ethylene-induced PFAZ (Smith and Whiting 2010). In the absence of a discrete PFAZ, the pedicel–fruit junction region in 'Chelan' corresponding to the PFAZ in 'Bing' and 'Skeena' was used for subsequent RNAseq analysis.

## 3.2. Transcriptome Assembly and Annotation

The transcriptome assembly resulted in the generation of 82,587 contigs from 1,061,563,488 total trimmed reads. Contigs were subsequently filtered for >200 base length and >2x coverage, for a final total of 81,852 contigs for downstream processing (Supplementary Materials Data S5). Functional annotation conducted using the OmicsBox genomics suite resulted in the assignation of annotations to a total of 30,946 (37.8%) contigs (Supplementary Materials Data S7).

## 3.3. Gene Ontology Enrichment Analysis of Differentially Expressed Contigs

Based on our NOIseq/Kal's test approach, 1274 genes were identified to be differentially expressed in ethephon-treated 'Bing', 715 in 'Skeena', and 523 in 'Chelan' in at least one treatment/timepoint in comparison with the control. It is to be noted that NOISeq uses a multinomial distribution to simulate replications. Therefore, the replicates are an approximation and show genes with changes over a set statistical threshold in the samples analyzed. The differentially expressed genes identified in this study thus represent potential activators of abscission zone formation [39].

To understand what biological processes, molecular functions, and cellular components were enriched across the three cultivars and thereby determine the pathways of greatest interest to this study, GO enrichment analysis of differentially expressed contigs was conducted, followed by filtering for the most specific ontologies using an FDR-corrected $p$-value cutoff of 0.05. This strategy resulted in the identification of enriched GOs at 6-h post-ethephon and at harvest for all genotypes. No GOs were significantly enriched at the 0-h time point.

### 3.3.1. Unique, Enriched Gene Ontologies at 6 Hours Post-Ethephon Treatment

From samples collected 6 h after treatment, 84 unique GOs were identified for 'Bing', 8 for 'Chelan', and 33 for 'Skeena' (Figures 2A and 3A). Among the ontologies unique to 'Bing' were several terms associated with chitin breakdown and metabolism, including: "chitinase activity", "chitin binding", "chitin catabolic process", and "response to chitin". Besides the hallmark role of plant chitinases and associated pathways in the breakdown of fungal pathogens, these enzymes are involved in a number of other important aspects of development, such as the generation of signal molecules that regulate organ morphogenesis [47], the regulation of lignin deposition and cell shape [48–50], regulation of programmed cell death [15], and the activation of abscission zones [15,51]. Chitin-associated processes have also been shown to be activated in response to jasmonic acid, methyl jasmonate, ethylene, and gibberellic acid [47,52,53], as well as auxin and cytokinin [54]. The unique enrichment of chitin-associated GO terms in 'Bing' was accompanied by the simultaneous overrepresentation of several hormone signaling pathways, including "methyl jasmonate esterase activity", "methyl salicylate esterase activity", "jasmonic acid catabolic process", "salicylic acid metabolic process", "gibberellin catabolic process", and "hormone-mediated signaling process", the latter of which included genes associated with ethylene, auxin, and abscisic acid (ABA) signaling and response. As jasmonic acid (JA) and salicylic acid (SA) metabolic pathways have established roles in plant growth regulation and stress response,

the unique co-enrichment of all of these terms, in conjunction with the "ROS response" enriched term, suggest that the inducible abscission phenotype may result from the unique jasmonic acid and redox/ROS signaling pathways, which are known to be activated in response to ethylene response [55]. These pathways, in turn, may lead to the activation of chitin-associated processes that lead to changes in lignin deposition, cell growth, and programmed cell death in a cascade of events leading up to abscission zone activation following ethephon application in 'Bing'.

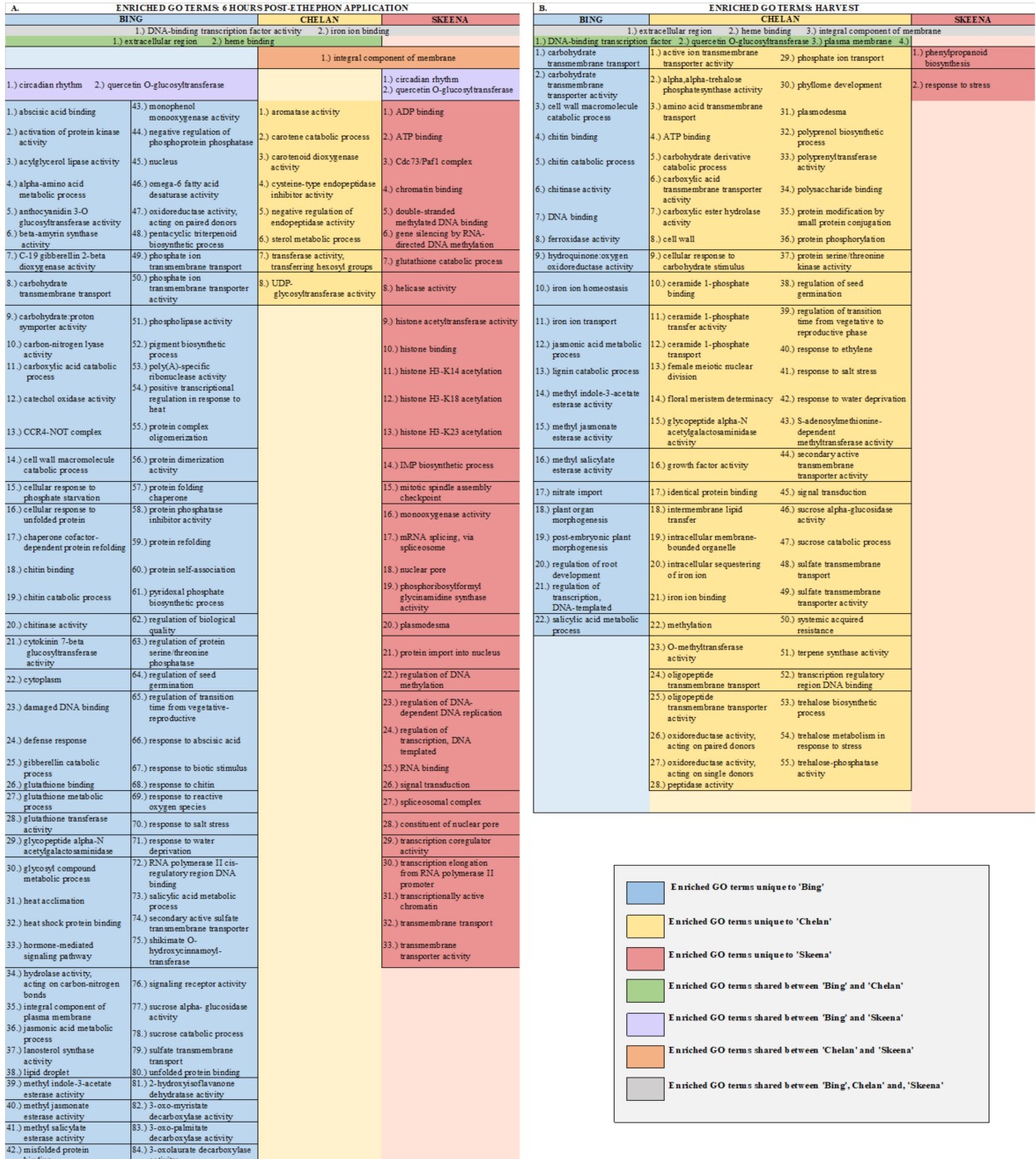

**Figure 2.** Chart displaying shared and unique enriched GO terms in ethephon-treated 'Bing', 'Chelan', and 'Skeena' at 6 h post-ethephon treatment (**A**) and at harvest (**B**). Enrichment results are based on Fisher's Exact test with an FDR corrected $p$-value < 0.05.

In 'Chelan', a relatively small number of GO terms were enriched at 6 h post-ethephon. Among the few enriched terms was "negative regulation of endopeptidase activity". As with plant chitinases, endopeptidases contribute to programmed cell death and cell wall breakdown [56] and have been shown to be activated by exogenous ethylene application [57]. The enrichment of this ontology suggests that molecular mechanisms are at play to stabilize the ethylene response in 'Chelan', whereas such a mechanism does not appear to play a substantial role in 'Bing'.

| Enriched GO Term (1) | Cultivar (2) | DE Genes Associated with Enriched GOs (3) | Contig (4) | Bing 0 Hr | Bing 6 Hr | Bing Harv | Chelan 0 Hr | Chelan 6 Hr | Chelan Harv | Skeena 0 Hr | Skeena 6 Hr | Skeena Harv |
|---|---|---|---|---|---|---|---|---|---|---|---|---|
| Chitinase activity/binding/catabolism; Chitin response | B | endochitinase 2-like | 22753 | * | * | | | | | | | |
| | B | endochitinase EP3-like v1 | 18291 | | * | | | | | | | |
| | B | endochitinase EP3-like v2 | 18292 | | * | | | | | | | |
| | B | endochitinase EP3-like v3 | 43552 | | * | | | | | | * | |
| | B | endochitinase-like | 3317 | | * | | | * | | | | |
| Hormone-mediated signaling pathway | B | abscisic acid receptor PYR1-like | 11946 | | * | | | | | | | |
| | B | ethylene receptor ETR2 | 6010 | | * | | | | | | | |
| | B | auxin responsive protein IAA1-like | 2474 | | * | | * | | | | | |
| | B | protein TIFY 5A-like | 44985 | | * | | | | | | * | |
| | B | protein TIFY 9 | 7749 | | * | | | | | | | |
| Jasmonic/salicylic acid metabolic process; Methyl IAA esterase activity | B | allene oxide cyclase, chloroplastic-like | 6411 | | * | | | | | | | |
| | B | 2-hydroxyisoflavanone dehydratase-like v1 | 17848 | | * | | | | * | | | |
| | B | 2-hydroxyisoflavanone dehydratase-like v2 | 17850 | | * | | | | * | | | |
| | B | 2-hydroxyisoflavanone dehydratase-like v3 | 31946 | | * | | | | | | | |
| | B | probable carboxylesterase 15 | 8498 | | * | | | | | | | |
| | B | probable carboxylesterase 18 | 9714 | | * | | | | | | | |
| | B | salicylic acid-binding protein 2-like v1 | 3684 | | * | | | | | | | |
| | B | salicylic acid-binding protein 2-like v2 | 3685 | | * | | | | | | | |
| | B | salicylic acid-binding protein 2-like v3 | 3686 | | * | * | | | | | | |
| | B | salicylic acid-binding protein 2-like v4 | 3687 | | * | * | | | | | | |
| | B | salicylic acid-binding protein 2-like v5 | 3688 | | * | | | | | | | |
| | B | salicylic acid-binding protein 2-like v6 | 3689 | | * | | | | | | | |
| | B | salicylic acid-binding protein 2-like v7 | 3690 | | * | * | | | | | | |
| | B | salicylic acid-binding protein 2-like v8 | 8874 | | * | * | | | | | | |
| | B | salicylic acid-binding protein 2-like v9 | 47678 | | * | * | | | | | | |
| ROS response | B | 16.9 kDa class I heat shock protein 1-like isoform X2 | 41037 | | * | | | | | | | |
| | B | 17.1 kDa class II heat shock protein-like | 14189 | | * | | | | | | | |
| | B | 17.4 kDa class III heat shock protein | 31787 | | * | | | | | | | |
| | B | 17.6 kDa class II heat shock protein-like v1 | 14190 | | * | | | | | | | |
| | B | 17.6 kDa class II heat shock protein-like v2 | 14191 | | * | | | | | | | |
| | B | 18.1 kDa class I heat shock protein-like v1 | 49189 | | * | | | | | | | |
| | B | 18.1 kDa class I heat shock protein-like v2 | 49819 | | * | | | | | | | |
| | B | 18.5 kDa class I heat shock protein v1 | 52677 | | * | | | | | | | |
| | B | 18.5 kDa class I heat shock protein v2 | 52677 | | * | | | | | | | |
| | B | 18.5 kDa class I heat shock protein v3 | 58968 | | * | * | | | | | | |
| | B | heat shock factor protein HSF30 isoform X1 | 7368 | | * | | | | | | | |
| Carotenoid dioxygenase activity; Carotene catabolic process | C | 9-cis-epoxycarotenoid dioxygenase 1 | 1910 | | * | | | * | | | | |
| | C | 9-cis-epoxycarotenoid dioxygenase | 1912 | | | | | * | | | | |
| Negative regulation of endopeptidase activity; Endopeptidase inhibitor activity | C | alpha-humulene 10-hydroxylase-like v1 | 11871 | | | | | * | | | | |
| | C | alpha-humulene 10-hydroxylase-like v2 | 11873 | | | | | * | | | | |
| | C | alpha-humulene 10-hydroxylase-like v3 | 16462 | | | | | * | | | | |
| | C | alpha-humulene 10-hydroxylase-like v4 | 16964 | | | | | * | | | | |
| Chromatin binding; Methylated DNA binding; Regulation of DNA methylation; Gene silencing by RNA-directed DNA methylation; Histone acetylation; Histone binding | S | nucleolar complex protein 3 homolog isoform X1 v1 | 3330 | | | | | | | | * | |
| | S | nucleolar complex protein 3 homolog isoform X1 v2 | 11292 | | | | | | | | * | |
| | S | polycomb group protein EMBRYONIC FLOWER 2 isoform X | 13007 | | | | | * | | | * | |
| | S | increased DNA methylation 1 v1 | 10315 | | | | | | | | * | |
| | S | increased DNA methylation 1 v2 | 17311 | | | | | | | | * | |
| | S | increased DNA methylation 1 v3 | 20390 | | | | | | | | * | |
| | S | histone acetyltransferase HAC1-like isoform X1 v1 | 944 | | | | | | | | * | |
| | S | histone acetyltransferase HAC1-like isoform X1 v2 | 19137 | | | | | | | | * | |
| | S | chromatin structure-remodeling complex protein SYD isoform | 396 | | | | | | | | * | |

**A. 6 HOURS POST-ETHEPHON** — LOW (Eth vs Ctrl Expression) ... Eth Exp ≈ Ctrl Exp ... HIGH (Eth vs Ctrl Expression)

**Figure 3.** *Cont.*

| Enriched GO Term (1) | Cultivar (2) | Description (3) | Contig (4) | Bing 0 Hr | Bing 6 Hr | Bing Harv | Chelan 0 Hr | Chelan 6 Hr | Chelan Harv | Skeena 0 Hr | Skeena 6 Hr | Skeena Harv |
|---|---|---|---|---|---|---|---|---|---|---|---|---|
| Chiting binding and catabolism; Chitinase activity | B | class V chitinase-like | 72624 | | | * | | | | | | |
| | B | endochitinase 2-like | 22753 | | * | * | | | | | | |
| | B | endochitinase EP3-like v1 | 16426 | | | * | | | | | | |
| | B | endochitinase EP3-like v2 | 16495 | | | * | | | | | | |
| | B | endochitinase EP3-like v3 | 16496 | | | * | | | * | | * | |
| | B | endochitinase EP3-like v4 | 6142 | | | * | | | | | | |
| Lignin catabolic process | B | laccase-15-like | 47488 | | | * | | | | | | |
| | B | laccase-7-like | 48560 | | | * | | | | | | |
| | B | putative laccase-9 v1 | 10596 | | | * | | | | | | |
| | B | putative laccase-9 v2 | 62006 | | | * | | | * | | | |
| Jasmonic acid/salicylic acid metabolic process | B | salicylic acid-binding protein 2-like v4 | 3687 | | * | * | | | | | | |
| | B | salicylic acid-binding protein 2-like v5 | 3688 | | * | * | | | | | | |
| | B | salicylic acid-binding protein 2-like v8 | 8874 | | * | * | | | | | | |
| | B | salicylic acid-binding protein 2-like v9 | 47678 | | * | * | | | | | | |
| Plant organ morphogenesis | B | LOB domain-containing protein 18 | 19670 | | | * | | | | | | |
| | B | NAC transcription factor 29-like | 11020 | | | * | | | | | | |
| | B | receptor-like protein kinase HSL1 | 2398 | | | * | | | | | | * |
| Cell wall | C | beta-galactosidase 8 | 2206 | | | | | | * | | | |
| | C | expansin-A1-like | 15072 | | | | | | * | | | |
| | C | pectinesterase-like | 21520 | | | | | | * | | | |
| | C | xyloglucan endotransglucosylase/hydrolase protein 32 | 46901 | | | | | | * | | | |
| | C | xyloglucan endotransglucosylase/hydrolase protein B | 26608 | | | | | | * | | | |
| Ethylene response | C | ethylene-responsive transcription factor ERF113-like | 1199 | | * | | | | * | | | |
| | C | respiratory burst oxidase homolog protein A | 7829 | | | | | | * | | | |
| | C | transcription factor MYB108-like | 24917 | | | | | | * | | | |
| | C | transcription factor MYBS3 | 13580 | | * | * | | | * | | | |
| Methylation | C | caffeic acid 3-O-methyltransferase-like v1 | 35033 | | | | | | * | | | |
| | C | caffeic acid 3-O-methyltransferase-like v2 | 36566 | | | | | * | * | | | |
| | C | methyltransferase-like protein 2 | 13169 | | | | | | * | | | |
| | C | methyltransferase At1g27930 | 36027 | | | * | | | * | | | |
| | C | salicylate carboxymethyltransferase-like | 12961 | | | | | | * | | * | |
| Phenylpropanoid biosynthetic process | S | fatty acyl-CoA reductase 3-like | 32869 | | | | | | | | | * |
| | S | phenylalanine ammonia-lyase 1 | 4716 | | * | | | | | | | * |
| Response to stress | S | 16.9 kDa class I heat shock protein 1-like isoform X2 | 41037 | | * | | | | | | | * |
| | S | defensin Ec-AMP-D2-like | 5976 | | | * | | | | | | * |
| | S | desiccation protectant protein Lea14 homolog | 8453 | | | | | | | | | * |
| | S | MLP-like protein 28 | 13433 | | | * | | | * | | | * |
| | S | MLP-like protein 43 | 1079 | | | * | | | * | | | * |
| | S | peroxidase 10 | 68269 | | | | | | | | | * |
| | S | peroxidase 17-like | 2517 | | | * | | | | | | * |

**Figure 3.** Heatmap displaying the expression of genes associated with significantly enriched, unique gene ontologies (GOs) for each genotype at 6 h post-ethephon treatment (**A**) and at harvest (**B**). Columns 1 and 2 display enriched GO terms and the genotype for which they were enriched, respectively. Columns 3 and 4 display the names of the differentially expressed genes associated with each enriched GO and their corresponding contig numbers, respectively. Heatmap represents fold change expression of ethephon-treated abscission zones (AZs) versus the control AZs for each cultivar. Asterisk indicates significant differential expression (Kal's Z-test FDR-corrected *p*-value < 0.05, NOISeq DE probability > 0.8, |Log2FC| > 1).

'Skeena' displayed a number of enriched ontologies associated with epigenetic modifications, including "chromatin binding", "double-stranded methylated DNA binding", "regulation of DNA methylation", "gene silencing by RNA-directed DNA methylation", "histone acetylation", and "histone binding". Such processes have previously been implicated in the abscission of citrus [58] and litchi [59]. In general, methylation of cell wall structural components, such as pectin, is thought to be critical for the maintenance of cell wall integrity [60]. Enrichment for such processes, which may occur in an ethylene-independent manner, could underlie the natural abscission process in 'Skeena'.

### 3.3.2. Unique, Enriched GOs at Harvest

At harvest, 22 unique enriched ontologies were identified in 'Bing', 56 in 'Chelan', and 2 in 'Skeena' (Figures 2B and 3B). Among the ontologies unique to 'Bing' were "lignin catabolic process" and "plant organ morphogenesis", along with several GO terms that

were also enriched at the 6-h time point: chitin-metabolism and response-associated terms and JA/SA-associated processes. Lignin, a derivative of phenylpropanoid metabolism, is abundant in abscission zones, aiding in the development of a protective boundary layer [19], serving as a mechanical brace to localize cell wall breakdown [61], and accumulating specifically in the PFAZ regions during ethylene-promoted abscission [17]. In addition to the instrumental role of lignin, general processes involved in "plant organ morphogenesis" appear to be at work in inducible abscission in 'Bing'.

In 'Chelan', a comparatively high number of terms was observed at harvest. Among the enriched GOs were "cell wall-related processes" and "methylation" (Figure 2B, Figure S9). These abscission-associated terms are enriched much later than they are for 'Bing' and 'Skeena', both of which showed the enrichment of associated processes at the 6-h time point. Furthermore, in contrast to the 6-h time point, at which few GOs were enriched, the elevated number of enriched terms observed in 'Chelan' at harvest suggest a delayed developmental response to ethylene, which may contribute to the failure of this cultivar to develop a PFAZ naturally and in the presence of ethephon.

In 'Skeena', at harvest, only two GO terms were uniquely enriched: "phenylpropanoid biosynthetic process" and "stress response". The limited number of unique terms at the later time point indicates that the majority of impactful processes contributing to PFAZ activation likely take place earlier in development. Among the enriched ontologies, phenylpropanoid metabolism is involved in the production of lignin precursors and has been previously implicated in fruit abscission [62]. Additionally, stress-associated processes appear to be active at this stage of development and are likely to further accentuate processes associated with programmed cell death, changes in cell wall integrity, and ultimately, PFAZ activation in 'Skeena'.

### 3.3.3. Shared, Enriched Gene Ontologies

In addition to unique GOs, shared ontologies reveal PFAZ activation-associated processes that are similar between two or more cultivars following ethephon treatment (Figure 2A,B and Figure 4). At harvest, 'Bing' and 'Chelan' shared the enriched GO term "response to auxin", while 'Skeena' did not. One possible contributing factor to the auto-abscission of 'Skeena' is that this cultivar may have naturally lower free auxin levels than 'Bing' or 'Chelan', promoting the natural activation of the PFAZ. This concept is explored further in a later section.

At the 6-h time point, all three genotypes shared the enriched GO term, "DNA-binding transcription factor activity". This GO term, among which a number of ethylene-responsive transcription factors were identified (Figure 4), remained enriched in 'Bing' and 'Chelan' at harvest. This observation provides evidence that all three cultivars have ethylene-responsive capacity; however, the previously described negative regulation of ethylene-associated mechanisms in 'Chelan', along with the enrichment of auxin response at harvest, suggests that a higher degree of feedback inhibition serves to quell the extent of ethylene-signaling and response that culminates in strong PFAZ activation in the other two cultivars (naturally in 'Skeena,' and following ethephon stimulation in 'Bing').

The term "integral component of membrane" was shared between 'Chelan' and 'Skeena' at 6 h post-ethephon and between all three cultivars at harvest. A large number of differentially expressed genes were associated with this term (Figure S9); among the notable genes were cell wall and integrity-associated *patatin-like protein 2* (*PLP2*), which plays a role in programmed cell death, and protein *walls-are-Thin 1* (*WAT1*), which, in addition to its role in auxin mobilization, plays a role in secondary cell wall formation and stability [63,64]. As all three cultivars undergo a decrease in PFRF following application of ethephon, it is logical that biological processes, molecular functions, and cellular components associated with membrane and cell wall integrity are affected (Figure 2).

**A. 6 HOURS POST-ETHEPHON**

LOW Eth vs Ctrl Expression — Eth Exp ≈ Ctrl Exp — HIGH Eth vs Ctrl Expression

| Enriched GO Term [1] | B | C | S | DE Genes Associated with Enriched GOs [3] | Contig # [4] |
|---|---|---|---|---|---|
| DNA-binding transcription factor activity | X | X | X | AP2/ERF and B3 domain-containing transcription factor R | 14058 |
| | X | X | X | AP2/ERF and B3 domain-containing transcription represso | 20340 |
| | X | X | X | ethylene-responsive transcription factor 2-like | 37379 |
| | X | X | X | ethylene-responsive transcription factor 4-like v1 | 22307 |
| | X | X | X | ethylene-responsive transcription factor ERF ABR1-like | 17992 |
| | X | X | X | ethylene-responsive transcription factor ERF027 | 39727 |
| | X | X | X | ethylene-responsive transcription factor ERF034-like | 30914 |
| | X | X | X | ethylene-responsive transcription factor ERF096-like | 68474 |
| | X | X | X | ethylene-responsive transcription factor ERF098-like | 64590 |
| | X | X | X | ethylene-responsive transcription factor ERF109-like v1 | 23882 |
| | X | X | X | ethylene-responsive transcription factor ERF109-like v4 | 41515 |
| | X | X | X | ethylene-responsive transcription factor ERF113-like | 1199 |
| | X | X | X | ethylene-responsive transcription factor ERF1B-like v1 | 47562 |
| | X | X | X | ethylene-responsive transcription factor ERF1B-like v2 | 50021 |
| | X | X | X | ethylene-responsive transcription factor ERF2-like | 20167 |
| | X | X | X | ethylene-responsive transcription factor TINY | 44704 |
| | X | X | X | ethylene-responsive transcription factor TINY-like | 54876 |
| | X | X | X | ethylene-responsive transcriptionfactor ERF027-like | 33777 |
| | X | X | X | transcription factor E2FB-like isoform X2 | 15691 |
| | X | X | X | CBF/DREB1-like protein a | 66067 |
| | X | X | X | transcription factor MYB114-like | 13853 |
| | X | X | X | MYB-related protein 308-like v2 | 75886 |
| | X | X | X | probable WRKY transcription factor 26 | 2506 |
| | X | X | X | probable WRKY transcription factor 33 | 7843 |
| | X | X | X | probable WRKY transcription factor 40 | 12849 |
| | X | X | X | probable WRKY transcription factor 69 | 30796 |
| | X | X | X | probable WRKY transcription factor 75 v1 | 22200 |

Cultivar [2] — Expression of Genes Associated with Enriched GOs (Bing, Chelan, Skeena; columns 0 Hr, 6 Hr, Harv)

**B. HARVEST**

LOW Eth vs Ctrl Expression — Eth Exp ≈ Ctrl Exp — HIGH Eth vs Ctrl Expression

| Enriched GO Term [1] | B | C | S | DE Genes Associated with Enriched GOs [3] | Contig # [4] |
|---|---|---|---|---|---|
| Auxin response | X | X | | auxin response factor ARF3 | 9103 |
| | X | X | | auxin-responsive protein IAA27 v1 | 864 |
| | X | X | | auxin-responsive protein IAA27 v2 | 8321 |
| | X | X | | auxin-responsive protein SAUR32 | 22530 |
| | X | X | | auxin-responsive protein SAUR50 | 41960 |
| | X | X | | Probable auxin efflux carrier component 1c | 5865 |
| | X | X | | protein BIG GRAIN 1-like A | 7489 |
| | X | X | | VAN3-binding protein-like isoform X1 | 8523 |
| DNA-binding transcription factor activity | X | X | | AP2/ERF and B3 domain-containing transcription represso | 20340 |
| | X | X | | ethylene-responsive transcription factor 4-like v1 | 22307 |
| | X | X | | ethylene-responsive transcription factor 4-like v2 | 17802 |
| | X | X | | ethylene-responsive transcription factor ERF017 | 31967 |
| | X | X | | ethylene-responsive transcription factor ERF023 | 31950 |
| | X | X | | ethylene-responsive transcription factor ERF027 | 39727 |
| | X | X | | ethylene-responsive transcription factor ERF034-like | 30914 |
| | X | X | | ethylene-responsive transcription factor ERF096-like | 68474 |
| | X | X | | ethylene-responsive transcription factor ERF109-like v1 | 23882 |
| | X | X | | ethylene-responsive transcription factor ERF109-like v2 | 41438 |
| | X | X | | ethylene-responsive transcription factor ERF109-like v3 | 23883 |
| | X | X | | ethylene-responsive transcription factor ERF109-like v5 | 58830 |
| | X | X | | ethylene-responsive transcription factor ERF1B-like v1 | 47562 |
| | X | X | | ethylene-responsive transcription factor ERF1B-like v2 | 50021 |
| | X | X | | MYB-related protein 308-like v1 | 40430 |
| | X | X | | MYB-related protein Myb4-like | 61117 |
| | X | X | | MYB10 V1-2 | 4548 |
| | X | X | | transcription factor MYB108-like | 24917 |
| | X | X | | transcription factor MYB36-like | 7229 |
| | X | X | | transcription factor MYB41-like | 60269 |
| | X | X | | transcription factor MYB8-like | 26167 |
| | X | X | | probable WRKY transcription factor 53 | 16663 |
| | X | X | | probable WRKY transcription factor 71 | 19579 |
| | X | X | | probable WRKY transcription factor 75 v2 | 31022 |
| | X | X | | transcription repressor MYB6-like | 13058 |

Cultivar [2] — Expression of Genes Associated with Enriched GOs (Bing, Chelan, Skeena; columns 0 Hr, 6 Hr, Harv)

**Figure 4.** Heatmaps displaying the expression of genes associated with significantly enriched gene ontologies (GOs) shared between two cultivars 6 h post-ethephon treatment (**A**) and at harvest (**B**). Columns 1 and 2 display enriched GO terms and the genotype for which they were enriched, respectively. Columns 3 and 4 display the names of the differentially expressed genes associated with each enriched GO and their corresponding contig numbers, respectively. Heatmap represents fold change expression of ethephon-treated abscission zones (AZs) versus the control AZs for each cultivar.

Overall, the GO enrichment results provide a global picture of the processes underlying the natural abscission capacity of 'Skeena', the inducibility of the abscission process in 'Bing' and recalcitrance to PFAZ activation in 'Chelan'. The examination of the differentially expressed genes associated with the enriched ontologies, described in the next section, lends further insight into the mechanisms underlying these three unique sweet cherry abscission phenotypes.

### 3.4. Differential Expression of Genes Associated with Key, Enriched Ontologies

The results of the GO enrichment analysis provided a basis for distinguishing pathways in which to observe the expression patterns of significant genes and to compare these patterns across cultivars following ethephon or control treatments.

#### 3.4.1. Differential Expression of Genes at 6 Hours Post-Ethephon Treatment

In 'Bing', a number of differentially expressed contigs corresponding to endochitinases, including several transcript variants of *endochitinase EP3-like*, *endochitinase-like*, and *endochitinase 2-like* were induced at the 6-h time point (Figure 3A). At the same time, hormone-mediated signaling genes showed heightened expression, including *abscisic acid receptor PYR1-like* (*PYR1-like*), *ethylene receptor 2* (*ETR2*), *auxin-responsive protein IAA1-like* (*IAA1-like*), and JA responsive protein*s TIFY5A-like* and *TIFY 9*. While genes associated with chitinase activity, hormone signaling, and ROS response displayed consistently elevated fold-change expression in 'Bing', in the other two cultivars, the responses varied. In general, a trend of low or negative fold change expression in ethephon-treated versus control PFAZ tissues was observed for this gene set in 'Chelan' and 'Skeena'; notably, however, *PYR1-like* and *IAA1-like* fold change expression was elevated in both 'Bing' and 'Chelan'. The high expression of ABA-associated *PYR1-like*, which promotes senescence and stress responses leading up to abscission, in the PFAZ of ethephon-treated 'Bing' and 'Chelan' cherries suggests that, despite ultimately different abscission phenotypes, both cultivars possess upstream mechanics that precede PFAZ activation. Furthermore, the low abscission capacity of 'Chelan' may lie in a comparatively high auxin versus ethylene response, along with negative regulatory mechanisms that have been previously undescribed. Specifically, several transcripts corresponding to *alpha-humulene 10-hydroxylase-like*, associated with the GO term "negative regulation of endopeptidase activity", were consistently downregulated in all three cultivars following ethephon treatment, but most drastically so in 'Chelan'. The specific role of alpha-humulene hydroxylase-like proteins, which modulate plant secondary metabolites (mainly sesquiterpenes) in response to herbivory, in abscission remains to be elucidated; however, the unique expression pattern in 'Chelan' points towards a potentially novel mechanism of negative regulation of abscission zone activation.

Many genes associated with epigenetic modification were differentially expressed in 'Skeena' at the 6-h time point, including three transcript variants corresponding to *increased DNA methylation 1* and two transcript variants corresponding to *histone acetyltransferase HAC1-like isoform X1* (*HAC1-like*) (Figure 3A). These genes were also induced in 'Bing' at the 6-h time point, but they were not significantly differentially expressed like in 'Skeena'. This finding suggests that methylation plays an important role in abscission, and that methylation and associated processes may occur via both ethylene-dependent ('Bing') and independent ('Skeena') pathways.

#### 3.4.2. Differential Expression of Genes at Harvest

Among the genes differentially expressed in 'Bing' at the harvest time point was "lignin catabolism"-associated *laccase 15-like, laccase 7-like, and laccase 9* (Figure 3B). The former, laccase genes, are involved in lignin synthesis and deposition; these processes are known to occur during AZ activation and are critical for lignin-promoted architectural changes at the cellular level [17,49]. The heightened expression of these genes observed uniquely in 'Bing' at harvest suggests that laccases, and therefore, structural modifications associated with lignin accumulation are active in the abscission process for this cultivar.

Also differentially expressed at harvest in 'Bing' was "plant organ morphogenesis"-associated *lateral boundary domain containing protein 18* (*LOB18*) (Figure 3B). *LOB18* has previously been implicated in boundary formation during root development. It is possible that this gene, or additional members of the LOB gene family, plays a similar role in boundary formation at the PFAZ [65,66]. *LOB8* expression was high at harvest in both 'Bing' and 'Skeena', but the expression difference between ethephon-treated and control PFAZ regions was significant only in the former. In addition to *LOB18*, a differentially expressed transcript corresponding to receptor-like protein kinase *HAESA-LIKE 1* (*HSL1*), which is also associated with the "plant organ morphogenesis" ontology (Figure 3B), was significantly downregulated in both 'Bing' and 'Skeena'. This is consistent with previous work in Arabidopsis that revealed decreased expression levels of HSL1 immediately prior to abscission; furthermore, loss of HSL1 function resulted in impeded floral and leaf abscission [67]. The differential expression of both *LOB18* and *HSL1* may underlie the ability to induce abscission in 'Bing' through boundary formation and the promotion of abscission-associated processes.

In addition to the genes that are directly involved in the modification of the PFAZ ultrastructure, the elevated expression of several transcript variants of corresponding to *salicylic acid binding protein 2-like*, which is associated with JA and SA metabolism, was observed at the harvest time point in 'Bing', indicating that JA signaling mechanisms remain active throughout development in 'Bing' following ethephon treatment. Similarly, the expression of JA-induced, chitinase-associated genes was also observed at the harvest.

In 'Chelan', several genes associated with cell wall modification, including *beta-galactosidase 8*, *pectinesterase-like*, and two *expansin* genes, displayed significantly heightened expression at harvest. These genes are involved in cell wall loosening, the degradation of linkages between structural components, and the digestion of cell walls and middle lamella in the AZ [29,68–70]. While cell wall modification genes displayed similar expression patterns in 'Bing' and 'Skeena', the extent of expression change between treatment and control was not significant for these cultivars. The expression patterns of several other genes in 'Chelan' indicate that this cultivar may have reduced lignin biosynthetic capacity in comparison to 'Bing' and 'Skeena'. While the expression of cell wall modifying associated genes suggests that 'Chelan' is partially competent to develop a PFAZ, it is possible that this cultivar does not possess some of the key molecular equipment to coordinate all of the associated processes, including lignification. This is evidenced by the observation of significantly decreased expression in a methylation-associated transcript corresponding to *caffeic acid 3-O-methyltransferase-like* (*CAOMT*) at harvest; conversely, this transcript displayed increased abundance in both 'Bing' and 'Skeena' at 6 h and at harvest. *CAOMT* lies upstream of the lignin biosynthetic process; decreased expression of this gene has been shown to lead to reduced lignification in poplar, and expression changes in *CAOMT-like* genes parallels PFAZ formation in citrus [17,71,72].

The expression patterns of genes corresponding to cell wall modifiers and lignin biosynthesis precursors that were identified in 'Chelan' lend insight regarding why this non-abscising cultivar displays reduced PFRF but fails to form a PFAZ that is discrete enough to facilitate mechanical harvesting; furthermore, low expression of other terminal abscission-associated genes, like the laccases displaying heightened expression in 'Bing' lend further credibility to this hypothesis.

Most notable among the differentially expressed genes in 'Skeena' were stress-response-associated *peroxidase 10*, *peroxidase 17-like*, and *16.9 kDa class I heat shock protein 1-like.* The elevated expression of stress-responsive genes indicates an increased need for ROS scavenging to prevent oxidative damage while these signal molecules serve the purpose of activating abscission-associated processes. Based on the pronounced decrease of stress-responsive genes in 'Skeena' following ethephon application and the observation that PFRF of 'Skeena' is reduced the least out of all the cultivars following treatment, it is plausible that 'Skeena' may only perceive the exogenous ethylene as environmental stress, whereas 'Bing' and 'Chelan' appear to host concerted, ethylene-induced signal transduction

responses upstream of abscission in response to ethephon application. In conjunction with increased peroxidase gene expression, reduced expression of pathogen defense-associated transcripts corresponding to *major latex protein* (*MLP-like protein*) *28* and *43* was observed. This is consistent with a study of *MLP-like* expression patterns in peach abscission zones, wherein it was suggested that *MLP-like* genes play a role in the mitigation in defense in early AZ formation but display reduced expression immediately prior to abscission and are further downregulated in response to ethylene [73]. Similar responses were observed for additional plant-defense-associated genes, including *dessication protectant protein Lea14* and *defensin Ec-AMP-D2-like*, in both 'Skeena' and 'Bing'.

*3.5. Differential Expression of Ethylene-Associated Contigs*

In addition to observing the expression patterns of differentially expressed contigs associated with key enriched metabolic pathways for each cultivar, it was of interest to specifically examine the expression of genes involved in ethylene response, to better understand how these responses may vary at the cultivar level and thereby, differentially contribute to abscission phenotypes. The basis for the inducibility of PFAZ activation in 'Bing' during fruit development is further evident at the genetic level with respect to the expression of ethylene biosynthesis, signaling, and responsive elements in the transcriptome (Figure S9). Throughout the time course in 'Bing', a transcript corresponding to the rate-limiting ethylene biosynthetic gene *1-aminocyclopropane-1-carboxylate oxidase 1* (*ACO1*) was highly elevated in expression, as were transcripts corresponding to *ethylene receptor 2* (*ETR2*) and *ethylene insensitive 2* (*EIN2*), two key genes involved in ethylene perception and signaling, respectively. The activation of ethylene biosynthesis that is observed to occur uniquely in 'Bing' suggests that exogenous ethephon stimulates the autocatalytic production of endogenous ethylene, which thereby accentuates the overall degree and impact of the ethylene response in this cultivar. Moreover, the naturally high expression of *ETR2* and *EIN2*, which are augmented in the presence of exogenous ethephon, lend further testament to the responsiveness of 'Bing' to ethylene. A negative regulator of ethylene, *reversion-to-ethylene-sensitivity 1* (*RTE1*), was induced in 'Bing' following ethephon application, but levels remained lower overall than in 'Chelan' and 'Skeena', an observation which is consistent with the comparatively high degree of sensitivity of 'Bing' to ethylene. In contrast to 'Bing', 'Skeena' and 'Chelan' displayed the basal levels of expression of *ACO1* and *ETR2* and elevated levels of *RTE1*, suggesting minimal ethylene response, endogenous production, and sensitivity (Figure 5, Figure S9).

The transmission of the ethylene-activated signal to the nucleus involves the ethylene-insensitive 2 (EIN2) and ethylene insensitive 2-like (EIL) family of proteins. Upon the activation of ETRs by ethylene, the C-terminal domain of EIN2 is cleaved and translocated to the nucleus where it activates the transcription of ethylene-responsive factors (ERFs) involved in the regulation of downstream ripening responses [74,75]. Numerous transcripts encoding ERFs were identified, which were differentially expressed in at least one cultivar and time point (Figure 4). Consistent with the aforementioned results, a majority of these contigs were highly induced in 'Bing' following ethephon treatment. In particular, transcripts corresponding to *ERF1B-like*, *ERF27* and *ERF109-like*, *ERF113-like*, *ERF109-like*, and *ERF ABR1-like* were highly differentially expressed in 'Bing' 6 h following ethephon treatment and also at harvest. In all three cultivars, *ERF96-like* was highly induced following ethephon treatment (Figure 4, Figure S9). In addition to ERFs, transcripts corresponding to members of the WRKY and MYB transcription factor families displayed highly elevated expression in 'Bing' following ethephon treatment. This is consistent with previous work demonstrating the activation of both of these transcription factor families during abscission, eliciting actions downstream of initial ethylene response [26,76–78].

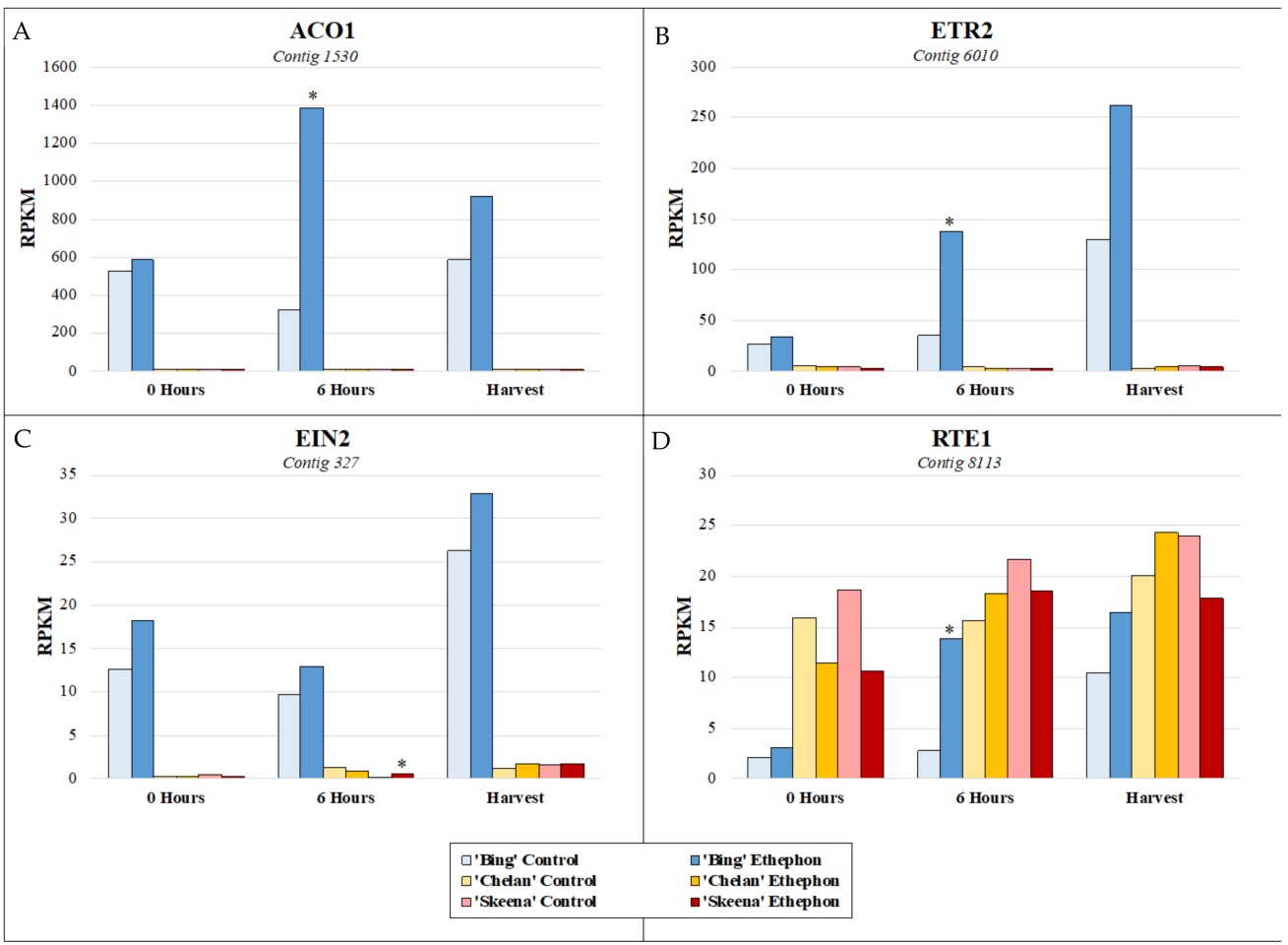

**Figure 5.** Expression patterns of ethylene biosynthesis gene (**A**) *1-aminocyclopropane-1-carboxylate oxidase 1* (*ACO1*), (**B**) *ethylene receptor 2* (*ETR2*), and (**C**) signal transducer *ethylene insensitive 2* (*EIN2*), and (**D**) ethylene-negative regulatory protein *reversion-to-ethylene-sensitivity 1* (*RTE1*) at 0 h, 6 h post-ethephon treatment, and harvest. Asterisks indicate significant difference, according to both Kal's Z-test ($p < 0.05$) and NOIseq-sim (probability > 0.8) differential expression probability analysis and a |logFC| > 1.

### 3.6. Differential Expression of Auxin Response-Associated Contigs

In addition to the differential ethylene responses, the ethephon-treated 'Chelan', 'Bing', and 'Skeena' PFAZ regions displayed a differential reduction in the expression of auxin-responsive genes in comparison to their respective controls. This suggests that the response to auxin and/or reduction in auxin transport was inhibited in the presence of exogenously applied ethylene.

Auxin-responsive transcription factor genes *IAA11*, *IAA13*, and *IAA27*, auxin transport facilitator *WAT1*, and auxin efflux carrier component 1c displayed comparatively low expression in 'Skeena', intermediate expression in 'Bing', and higher expression in 'Chelan' control PFAZ tissues at 6 h post-ethephon and at harvest. Ethephon treatment attenuated this expression in the PFAZ region for all three cultivars, but the same relative expression trend was maintained. *IAA13*, *IAA27*, and *WAT1* displayed decreased expression at harvest in all ethephon-treated cultivars in comparison with their respective controls. Interestingly, 'Chelan' displayed a natural increase in the expression of these transcripts from the 0-h time point to harvest, which was inhibited and reversed by the application of ethephon (Figure 6). For both 'Bing' and 'Skeena', the same transcripts naturally decreased in abundance over time, a decrease that was accelerated in the presence of ethephon. This finding is consistent with studies of fruitlet abscission in mango and other fruit, which describe a reduction in the transport of auxin through the pedicel and, in conjunction with increased

sensitivity for ethylene in the pedicel abscission zone, resulting in the induction of fruitlet abscission [26,27].

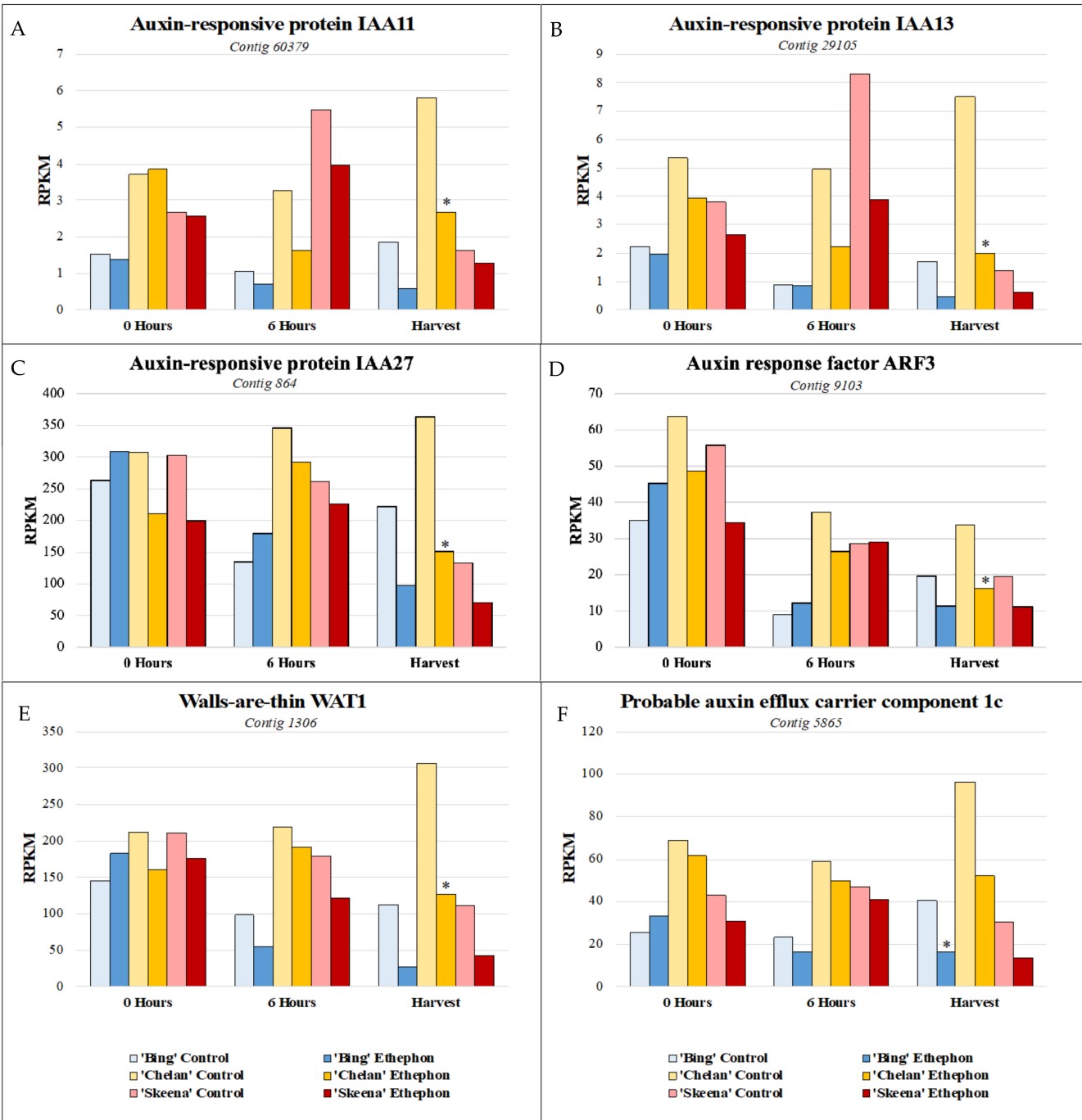

**Figure 6.** Expression patterns of auxin-associated genes. (**A**) shared expression pattern was observed for auxin-associated genes at harvest. Auxin-responsive transcription factors *IAA11*, *IAA13*, and *IAA27* (**A**–**C**), auxin transport facilitator *walls-are-thin 1* (*WAT1*) (**E**), and auxin efflux carrier component 1c (**F**) displayed comparatively lower expression in 'Skeena', intermediate expression in 'Bing', and higher expression in 'Chelan' control at harvest. Following ethephon treatment, expression was attenuated in all three cultivars; however, the same relative expression trend was maintained, with 'Skeena' ethephon-treated fruit exhibiting lowest expression at harvest and 'Chelan' ethephon-treated fruit displaying the highest expression at harvest. Asterisks indicate significant difference, according to both Kal's Z-test ($p < 0.05$) and NOIseq-sim differential expression probability analysis (probability > 0.8), and a |logFC| > 1.

While free auxin was not measured in this study, differences in the abundance of auxin-responsive and auxin mobilization-associated transcripts in the three sweet cherry cultivars over time lends to the extrapolation of information regarding cultivar-specific, endogenous free IAA concentrations. If free auxin levels are high, abundance and/or activity of auxin-responsive protein encoding transcripts (ARFs, IAAs) is expected to be higher to accommodate them. 'Skeena' does not require ethylene for abscission; however, ethephon application appeared to further offset the auxin to ethylene ratios in favor of ethylene to accelerate abscission in this cultivar. The ability of 'Skeena' to auto-abscise suggests that the endogenous accumulation of free auxin at the site of abscission may be naturally lower in this cultivar than it is for 'Bing' and 'Chelan'. Conversely, the comparatively higher expression of transcripts associated with auxin response and movement proteins in 'Chelan' is suggestive of a naturally higher level of endogenous IAA. While the current industrial standard levels of ethephon application were insufficient to reduce the PFRF of 'Chelan' fruit to the threshold required for mechanical harvesting, both PFRF and auxin-associated transcript abundance in 'Chelan' did decrease as a result of ethephon application. Ethephon application rates as high as 5.8 L ha$^{-1}$ remain insufficient to induce a reduction of 'Chelan' PFRF values to the threshold for mechanical harvest [10]. An early study found that a high application rate of 500 ppm (7.8 L ha$^{-1}$ [6.7 pt A$^{-1}$]), ethephon begins to deleteriously affect some sweet cherry cultivars by inducing unwanted leaf abscission and terminal shoot necrosis, although it is not reported to what extent 'Chelan' is impacted by such a concentration [10,11]. Considering the present results alongside the insight gained from previous work, it is possible that the mechanical abscission of 'Chelan' could be achieved through a combination of a slightly higher ethephon application rate and the application of auxin inhibitors, the latter of which could further shift the ethylene/auxin ratio in a manner favorable to abscission while reducing the need for excessively high and potentially phytotoxic ethephon application rates.

'Bing' PFAZ tissues displayed an abundance of auxin-associated transcripts intermediate to that of 'Skeena' and 'Chelan'. Following ethephon treatment, the phenotypic observations of inducible abscission were supported at the gene expression level, with the responses greatly attenuated to lower than those of 'Skeena' ethephon-treated and control PFAZ tissues.

### 3.7. Hypothetical Models for PFAZ Activation

Together, the GO enrichment and differential expression analysis results inform new, hypothetical models for abscission (Figure 7). In 'Bing', ethephon elicits an ethylene-response that stimulates hormonal regulatory signals and metabolic processes, including JA- and SA-mediated signaling. These processes, in turn, activate downstream processes associated with morphological changes at the PFAZ, including chitin metabolism and lignin biosynthesis and deposition. The culmination of these events ultimately facilitates the mechanical separation of the fruit at the PFAZ. The components of this process, which appears complete in 'Bing', seem to be partially lacking in 'Chelan', perhaps due to inherently high auxin levels that limit the ethylene-response capacity from the beginning. 'Skeena' appears to orchestrate PFAZ activation via a different and ethylene-independent mechanism. There is some recent evidence of an ethylene-independent abscission [79]. While this mechanism will require further elucidation, the lack of enrichment for the "auxin response" ontology and the reduced expression of the corresponding auxin-associated genes in 'Skeena' suggests that low levels of free auxin and therefore, the minimal antagonism of a natural abscission process may underlie the auto-abscission phenotype. This model lays foundations for future work building upon the transcriptomic results and validating the proposed mechanisms of abscission in each genotype, including analysis of lignin deposition, ROS, programmed cell death, and epigenetic reprogramming.

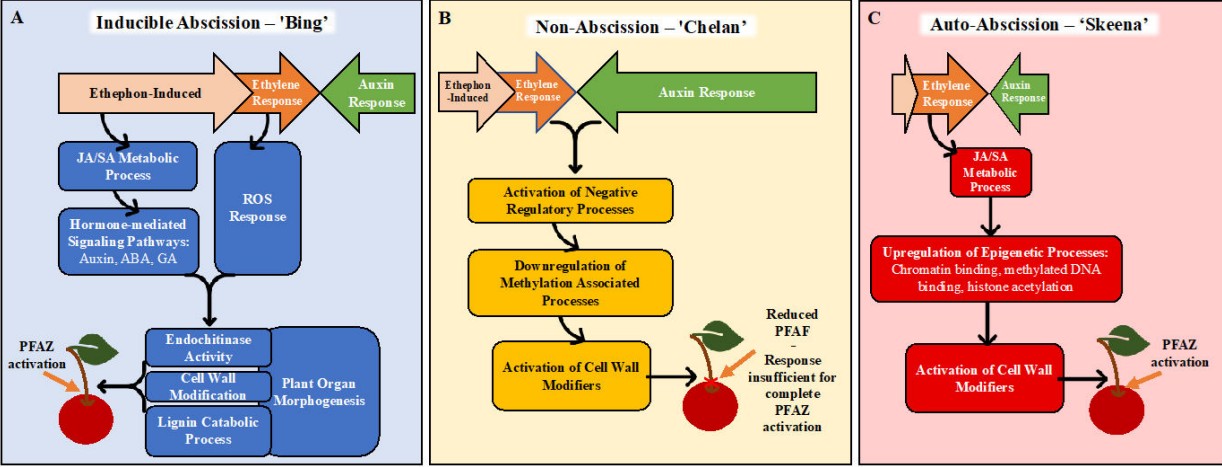

**Figure 7.** Hypothetical models for abscission, based on gene expression and gene ontology (GO) enrichment results following ethephon application of 'Bing', 'Chelan', and 'Skeena' sweet cherry. In 'Bing' (**A**), ethephon offsets the balance between ethylene and auxin in favor of ethylene. This triggers endogenous ethylene-responses that stimulates JA- and SA-mediated signaling, additional hormone-mediated signaling pathways and reactive oxygen species (ROS) responses. Together, these ethylene-induced signaling pathways activate downstream processes associated with morphological changes at the PFAZ, including chitin metabolism, cell wall modification, plant organ morphogenesis, and lignin biosynthesis and deposition. The culmination of these events ultimately facilitates the mechanical separation of the fruit at the PFAZ. In 'Chelan' (**B**), the components required for inducible abscission seem to be partially lacking. While ethephon likely contributes to enhanced ethylene signaling, free auxin levels in 'Chelan' may be high enough initially to overwhelm any ethylene response, thus limiting the cascade of signaling events leading to abscission. The activation of cell-wall-modifying enzymes may be responsible for the observed reduction in PFRF; however, it does not seem to be enough to facilitate complete PFAZ activation. In 'Skeena' (**C**), ethephon does little to affect the overall abscission phenotype, as this cultivar is capable of natural abscission. While ethylene signaling is not high, low initial levels of free auxin may naturally shift the hormonal tug-of-war in favor of abscission, leading to the moderate activation of JA metabolism and signaling. Near the time of harvest, upregulation in processes associated with epigenetic modification, and, ultimately, the activation of cell wall modifiers may represent intermediates in an alternate, ethylene-independent pathway to PFAZ activation.

### 3.8. Short Variant Discovery in Differentially Expressed Genes

Variant calling using the GATK pipeline revealed several SNPs and indels in the ORFs of DE genes associated with key enriched GOs (Table 1).

An INDEL was identified in "DNA-binding transcription factor activity"-associated *AP2/ERF and B3 domain containing protein RAV1*, wherein the deletion of a 6 bp segment results in the loss of two amino acid residues without impacting the reading frame. Interestingly, 'Skeena' was homozygous for this deletion; 'Bing' was heterozygous, and 'Chelan' did not possess the deletion. The resulting protein changes likely impact the natural propensity of the cultivars to transduce signals associated with stress and hormone responses upstream of PFAZ activation and could ultimately underlie differences in abscission phenotype. Additionally, an SNP, resulting in an amino acid substitution in the coding sequence of *auxin efflux carrier component 1c*, was identified in 'Chelan'. This variant could impact the overall auxin responsiveness, which appears to be greater in 'Chelan' than 'Bing' or 'Skeena', at the genetic level.

Additional variants identified include a SNP in "ROS response"-associated *17.4 kDA class III heat shock protein*, an INDEL in the epigenetic modification-associated gene *polycomb group EMBRYONIC FLOWER 2 isoform x1,* and an SNP in the 3′ UTR of the "chitinase activity"-associated *endochitinase-like* gene. A member of the diverse polycomb group (PcG) family, *EMF2* is involved in the epigenetic silencing of developmental repressors and was significantly downregulated in 'Skeena' at the 6-h time point. While further work is needed to determine the precise nature of these variants and their impact on PFAZ activation, their presence in DE genes associated with enriched functions following

ethephon application makes them key targets for understanding the activation of abscission in sweet cherry cultivars.

**Table 1.** Variants identified in differentially expressed genes. The GATK RNAseq short variant discovery pipeline was used for the characterization of SNPs and indels [45]. Information regarding the associated enriched gene ontology (GO) terms, variant position, full protein length, residue number, type of variant, and whether the variants identified result in a change in amino acid sequence or open reading frame (ORF) is shown. Asterisks reference overlapping deletions.

| Gene | Associated Enriched GOs | Variant Position on Contig (bp) | Full Protein Length (Residues) | Variant Residue Number | Variant Type | Amino Acid/ORF Change | Genotype | | |
|---|---|---|---|---|---|---|---|---|---|
| | | | | | | | **Bing** | **Chelan** | **Skeena** |
| *Endochitinase-like* Contig 3317 (1140 bp) | Chitinase activity/binding/catabolism; Chitin response | 38 | 316 <br> Translation −2 | In 3′ UTR | SNP: A↔G | N/A | A/G | A/G | G/G |
| *Polycomb group protein EMBRYONIC FLOWER 2 isoform X1* Contig 13007 (913 bp) | Chromatin binding; Methylated DNA binding; Regulation of DNA methylation; Gene silencing by RNA-directed DNA methylation; Histone acetylation; Histone binding | 724 | 577 <br> Translation −1 | 453 | DEL: ACTTGTTC-CAATGT-CATTGG* | Frame shift impacting CDS | DEL/* | */* | */* |
| *17.4 kDa class III heat shock protein* Contig 31787 (1075 bp) | ROS Response | 747 | 161 <br> Translation +1 | 52 | DEL: CCACTG-GAATGC-TAGC-CGCTC-CTTTGTTCTCGTTGTTG-GTTTCGTGGGCC* | Frame shift resulting in early STOP | */* | */* | DEL/DEL |
| *Probable auxin efflux carrier component 1c* Contig 5865 (2848 bp) | Auxin response | 1770 | 619 <br> Translation +1 | 341 | SNP: G↔C | Glu↔Asp | G/G | C/C | G/G |
| *AP2/ERF and B3 domain-containing transcription factor RAV1* Contig 14058 (1503 bp) | DNA-binding transcription factor activity | 363 | 422 <br> Translation +2 | 353 | DEL: CGGCAT* | Gly-Ile deletion; reading frame unaffected | */DEL | */* | DEL/DEL |

*3.9. RT-qPCR Validation*

RT-qPCR analysis of 10 ethylene-responsive/abscission-related genes resulted in 70% correspondence of general expression trends for 'Bing' and 'Chelan' and an 80% correspondence for 'Skeena'. Validated transcripts whose RT-qPCR expression pattern (fold-change calculated using the $2^{-(\Delta\Delta Ct)}$ method) was consistent with that of the RNAseq-based expression (ethephon/control RPKM ratio) included genes associated with ethylene biosynthesis (ACS1 and ACO1), perception (ETR2), and response (ERF1B-like, ERF027-like, and WRKY1), as well as cell wall breakdown-associated polygalacturonase (PG) (Supplementary Materials Data S8).

**4. Conclusions**

This study investigated the changes in PFRF, gene expression changes, and enriched biological processes underlying PFAZ activation in sweet cherry. The results provide transcriptomic insight regarding the gene-expression-level effects of exogenous ethylene application on the abscission phenotype and PFAZ development, which can be utilized by the industry to customize harvest strategies.

Consistent with previous work, PFAZ activation was observed to occur in a cultivar-specific manner, and the abscission phenotype of each cultivar was affected to a different degree by the exogenous application of ethylene. The observation of unique, heightened expression of ethylene biosynthesis, perception, signaling, and response genes in 'Bing' at

6 h following ethephon application and at harvest parallels the decrease in PFRF and may be partially responsible for the inducibility of abscission in this cultivar. GO enrichment analysis revealed key biochemical pathways containing additional differentially expressed genes potentially underlying the unique abscission responses of each cultivar. Among these enriched pathways were a number of hormone-signaling pathways, ROS signaling, chitin metabolic processes, lignin biosynthesis, and DNA-binding transcription factor activity (among which were numerous ERFs).

Overall, the results of this study point towards a potential genetic basis for the inducible abscission response in 'Bing', the auto-abscission of 'Skeena', and the recalcitrance to abscise displayed by 'Chelan'. The natural abscission of 'Skeena', which is enhanced by ethephon, may result from naturally lower levels of free auxin, as evidenced by the low abundance of IAA/ARF-associated transcripts in comparison with 'Bing' and 'Chelan'. Furthermore, 'Skeena' exhibited fewer significant gene expression changes than the other cultivars following ethephon application, which corresponded to fewer enriched ontologies at the time of harvest. In 'Chelan', the comparatively high abundance of auxin-associated transcripts in control fruit, which was attenuated following ethephon treatment, may indicate higher levels of endogenous free auxin. The increased capacity for auxin mobilization, indicated by IAA/ARF expression, antagonizes the abscission-promoting effects of exogenous ethylene. This observation, along with the PFRF results, provides insight regarding the recalcitrance of 'Chelan' fruit to abscise at the pedicel–fruit junction.

The identification of cultivar-specific, differentially expressed genes and enriched GO terms involved in abscission, as well as ontologies that are shared across cultivars, provides information that will inform future efforts to promote the controlled, timely abscission of sweet cherries. This, in turn, will lead to the improvement and standardization of mechanical harvesting, thereby improving efficiency and increasing the economic profitability of the sweet cherry industry. Ultimately, the outcomes of this work may be extended to other crops where planned abscission can be useful in managing the harvest.

**Supplementary Materials:** The following are available online at https://www.mdpi.com/article/10.3390/horticulturae7090270/s1, Figure S1: Ethephon treatment and sampling timeline; Data S2. PFAZ data from 2010, 2013, and 2014 seasons; Data S3: Supplemental fruit color data collected at the same time as PFRF and AZ tissue sampling; Figure S4: Picture of fruit section sampled for PFAZ analysis; Data S5: Cherry assembly fasta file; Data S6: Spreadsheet with RPKM values of contigs that passed stringent statistical and probability filters for differential expression based on Kal's Z-test, NOIseq-sim analysis, and Log2FC expression filtering; Data S7: List of all annotated contigs >2 × coverage and >200 bp in length; Data S8: RT-qPCR primers; Cq values and fold change expression summary by template; Figure S9: Expression heatmaps for differentially expressed (DE) genes associated with shared and unique enriched gene ontology (GO) terms and DE genes associated with ethylene and auxin pathways.

**Author Contributions:** Conceptualization, A.D. and M.W.; methodology, B.K., T.K., M.W., and A.D.; formal analysis, S.H., B.K., and A.D.; investigation, S.H., B.K., T.K., J.A., and A.D.; resources, M.W. and A.D.; data curation, S.H. and A.D.; writing—original draft preparation, S.H., B.K., and A.D.; writing—S.H. and A.D. All authors have read and agreed to the published version of the manuscript.

**Funding:** This study was supported by a USDA Special Crop Research Initiative (SCRI) program grant (Project No. 2009-02559) to MW and AD. Work in the Dhingra lab was supported in part by Washington State University Agriculture Center Research Hatch Grant WNP00011. SLH, BK, and TK acknowledge the support received from ARCS Seattle Chapter and National Institute of Health/National Institute of General Medical Sciences through an institutional training grant award T32-GM008336.

**Institutional Review Board Statement:** Not applicable.

**Informed Consent Statement:** Not applicable.

**Data Availability Statement:** All sequence data is available from the NCBI Short Read Archive—'Bing' accession: SRX2210365; 'Chelan' accession: SRX2210366; 'Skeena' accession: SRX2210367

submitted under BioProject PRJNA329134. This manuscript has been released as a pre-print at BioRxiv [80].

**Acknowledgments:** The authors thank Richard Sharpe for assistance with RT-qPCR assays and analysis.

**Conflicts of Interest:** The authors declare no conflict of interest. The funders had no role in the design of the study; in the collection, analyses, or interpretation of data; in the writing of the manuscript, or in the decision to publish the results. The contents of this work are solely the responsibility of the authors and do not necessarily represent the official views of the funders.

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
