# Peer review of "Transcriptome Analysis Reveals Potential Mechanisms for Ethylene-Inducible Pedicel–Fruit Abscission Zone Activation in Non-Climacteric Sweet Cherry (Prunus avium L.)"

_horticulturae, doi:10.3390/horticulturae7090270_

Round 1
Reviewer 1 Report
In their paper, Hewitt et al, describe the analysis of the transcriptome of three sweet cherry cultivars that differ in the formation of an abscission zone at the fruit-pedicel junction. After measuring the pedicel-fruit retention force to certify the differences in the pedicel-fruit abscission zone and following treatment of the three cultivars with ethephon, the authors performed a detailed RNA-seq analysis in these zones where they find numerous differences in gene ontology (GO) enrichment pathways and differentially expressed genes. Based on these analyses, they present hypothetical models for abscission in the three cultivars that summarize their conclusions on the genetic (and epigenetic) basis of abscission zone control and the importance of hormone balancing.
The manuscript is well-written and balanced, the analysis is detailed, conclusions are supported by results. The findings of this work are important and provide a valuable insight for the customization of harvesting strategies in sweet cherries and for the development of cultivars in the future.
There is only one minor remark. Why did the authors choose to harvest 10 fruits per cultivar per time point as their biological replicates and then homogenize them into a single RNA-sequencing sample? Wouldn't be more sound if they performed triplicates analysis by homogenizing the ten fruits in three different biological replicates and then sequence them all of them?
Author Response
Ethephon application was done at 80% fruit maturity. However, due to the position of the fruit on the tree, variation in the fruit development stages is expected. To address the variability issue and the availability of resources, the approach of sampling the pedicel fruit abscission zone from 10 fruit was preferred to ensure a broader representation of the fruit in a sequenced sample. To gain additional confidence in the observations qRT-PCR analysis of several genes was included to complement and validate the RNAseq analysis.
Reviewer 2 Report
- The topic of the work is relevant. By providing localization of ROS across three Prunus avium genomes, the work would only benefit.
- It is also desirable to show changes in the deposition of lignin by a qualitative method..
- Epigenetic modifications are of interest, but the results need to be described in more detail.
.
It is desirable that the programmed cell death be supported by other (qualitative) methods.
- Fig. 2, 3, 4 are poorly readable, it is necessary to increase the font size.
Author Response
The reviewer has astutely identified several follow-on studies from the transcriptome analysis of the observed differences in the abscission phenotype of three sweet cherry cultivars presented in this manuscript. However, the proposed experiments were outside the scope of the present study.
To address the reviewer’s feedback, we have added a statement at lines 716-719 explaining that the topics mentioned above should be the subject of future research to evaluate the proposed model presented in Figure 7.
As advised, the font size of Figures 2, 3, and 4 has been increased.